# RadAgents: Multimodal Agentic Reasoning for Chest X-ray Interpretation with Radiologist-like Workflows

**Kai Zhang**[*,1,2] (iD)                      KAZ321@LEHIGH.EDU
**Corey D Barrett**[1]                 COREY.BARRETT@ORACLE.COM
**Jangwon Kim**[1]                  JANGWON.KIM@ORACLE.COM
**Lichao Sun**[2]                       LIS221@LEHIGH.EDU
**Tara Taghavi**[1]                   TARA.TAGHAVI@ORACLE.COM
**Krishnaram Kenthapadi**[1]        KRISHNARAM.KENTHAPADI@ORACLE.COM
[1] *Oracle Health AI*
[2] *Lehigh University*

**Editors:** Accepted for publication at MIDL 2026

## Abstract

Agentic systems offer a potential path to solve complex clinical tasks through collaboration among specialized agents, augmented by tool use and external knowledge bases. Nevertheless, for chest X-ray (CXR) interpretation, prevailing methods remain limited: (i) reasoning is frequently neither clinically interpretable nor aligned with guidelines, reflecting mere aggregation of tool outputs; (ii) multimodal evidence is insufficiently fused, yielding text-only rationales that are not visually grounded; and (iii) systems rarely detect or resolve cross-tool inconsistencies and provide no principled verification mechanisms. To bridge the above gaps, we present **RadAgents**, a multi-agent framework that couples clinical priors with task-aware multimodal reasoning and encodes a radiologist-style workflow into a modular, auditable pipeline. In addition, we integrate grounding and multimodal retrieval-augmentation to verify and resolve context conflicts, resulting in outputs that are more reliable, transparent, and consistent with clinical practice.

**Keywords:** Multi-agent system, multimodal reasoning, chest X-ray, image interpretation.

## 1. Introduction

Chest X-ray (CXR) imaging is a cornerstone of pulmonary screening, diagnosis, and follow-up, accounting for the largest share of diagnostic radiology examinations performed worldwide each year (Cid et al., 2024). Yet systematic assessment of thoracic structures remains labor-intensive, imposing a substantial time burden on radiologists (Fallahpour et al., 2025). The gradual introduction of AI into clinical practice shows promise for alleviating this workload (Zhang et al., 2024; Tanno et al., 2025). However, prevailing systems fall short on complex multimodal reasoning, such as integrating findings across disparate image regions, views, and time points, which is central to radiologists' practice. Most methods adhere to end-to-end designs in which the visual encoder performs a *single front-end pass* and subsequent reasoning proceeds *purely in text* (Wang et al., 2025). This encode-once, text-only paradigm decouples the reasoning trajectory from evolving visual evidence, leading to failures on tasks that require iterative re-inspection, precise measurements, and cross-comparisons (Liu et al., 2025) as shown in Figure 1.

---

[*] Internship at Oracle Health AI.

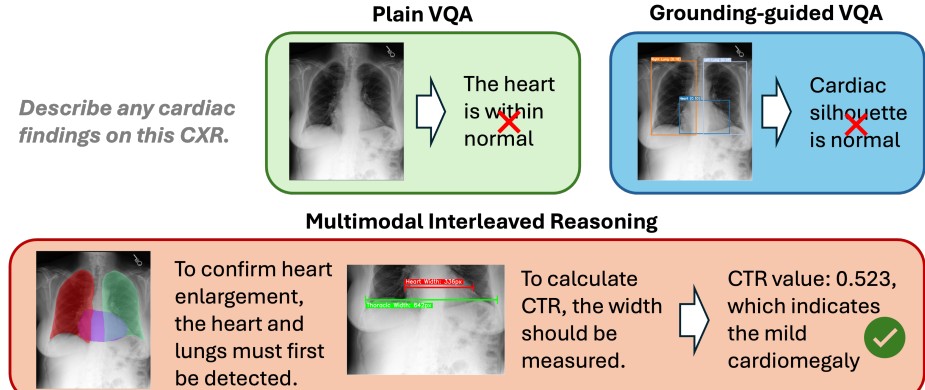

Figure 1: Prior "encode-once, text-only" (left) and grounding/cropping-only variants (right) can fail on queries requiring iterative re-inspection and quantitative assessment (e.g., cardiothoracic ratio). RadAgents instead supports multimodal interleaved reasoning, where hypotheses trigger targeted visual operations and the final answer is grounded in explicit visual evidence.

A promising direction is to *augment* large language models, including multimodal variants, with *external tools* (Lu et al., 2025). By delegating perceptual and classification subtasks such as organ or region segmentation and disease classification to validated modules, the language model can focus on planning and synthesis. Several agentic frameworks have explored this idea, ranging from training small models for limited tool use (Li et al., 2024; Nath et al., 2025) to pipeline systems that invoke general-purpose models for more flexible operations (Jiang et al., 2025; Schmidgall et al., 2024). In CXR interpretation, RadFabric (Chen et al., 2025) integrates diagnostic agents with a separate reasoning agent, and MedRAX (Fallahpour et al., 2025) expands task coverage by incorporating additional task-specific models. Despite improvements over single-model baselines, these systems often remain opaque and weakly aligned with clinical workflow: integration and reasoning steps are not explicitly traceable, visual and textual evidence are loosely coupled, and inconsistencies across tools are not systematically detected or resolved, which undermines trust and creates safety risks.

By contrast, radiologists reason through structured, radiology-specific workflows. For CXR, training and guidelines emphasize systematic review schemes, explicit quantitative assessments (e.g., cardiothoracic ratio, carinal angle), and comparison across time and views (Hodler et al., 2019). This process is inherently *interleaved*: clinicians move back and forth between image inspection, measurements, and textual synthesis, refining hypotheses as new evidence is obtained. Crucially, the reasoning is explicit and traceable, enabling peer review, error analysis, and integration with broader clinical context. Current multimodal LLM systems capture this paradigm only implicitly: reasoning is buried inside the model, tool calls are ad hoc, and there is limited support for auditing *how* a conclusion was reached or *which* intermediate steps failed (Lee et al., 2025).

To bridge this gap, we present **RadAgents**, a multi-agent framework for complex multimodal reasoning in CXR that encodes radiologist-like workflows into a modular, auditable pipeline. RadAgents decomposes interpretation into seven specialized agents: five subagents that implement core radiologic review modes, an *Orchestrator* that analyzes queries and dispatches tasks, and a *Synthesizer* that aggregates outputs, performs contextual verification, and resolves cross-tool conflicts. Each subagent follows predefined, radiologist-style workflows when applicable (e.g., for cardiomegaly or pleural effusion), while out-of-template queries fall back to workflow-free ReAct-style reasoning. Throughout, RadAgents maintains explicit logs of intermediate artifacts (segmentations, measurements, retrieved exemplars, and rationales), yielding step-level traceability instead of a single opaque explanation.

From a deployment standpoint, we instantiate RadAgents with open-source, lightweight vision–language models (Qwen3-VL-Instruct 4B/8B/30B (Bai et al., 2025)) as the core engines of each agent. We show that, when paired with structured workflows, tool integration, and conflict resolution, an 8B open model can match or surpass GPT-4o and specialist CXR models such as CheXagent (Chen et al., 2024) on diverse benchmarks. This suggests that trustworthy, workflow-aligned CXR reasoning does not strictly require very large proprietary models and can be realized with more accessible, on-premise-friendly architectures. Our contributions are threefold:

- We formalize *radiologist-like multimodal workflows* for CXR interpretation and encode them in a multi-agent system that interleaves visual evidence, measurements, and textual reasoning. RadAgents supports both workflow-guided and workflow-free modes, enabling coverage of common guideline-style tasks as well as open-ended queries.

- We design a *traceable, tool-augmented agentic architecture* that combines an Orchestrator, subagents, and a Synthesizer with retrieval-augmented conflict resolution and short-term memory. This yields explicit step-by-step trajectories and principled handling of cross-tool inconsistencies, improving alignment with clinical practice.

- We conduct an extensive study on across three challenging multimodal medical-reasoning datasets. RadAgents consistently outperforms competitive baselines by a substantial margin, and ablations show the importance of radiologist-like workflows, visual retrieval, and targeted scaling of the Synthesizer.

## 2. Methodology

RadAgents is a multi agent system with seven specialized agents (Figure 2). Five implement the clinical **ABCDE** review scheme (Hodler et al., 2019): **A**irway, **B**reathing, **C**irculation, **D**iaphragm, and **E**verything else. In addition, an *Orchestrator* agent analyzes each query and routes tasks to the appropriate specialists with the required patient context (for example, imaging view and prior studies), and a *Synthesizer* agent integrates their outputs, resolves conflicts, and produces the final output. This design confines context to task specific compartments, reducing the information each agent must process and simplifying context compression by having each sub-agent produce an initial summary for downstream synthesis. It also allows parallel execution, lowering latency for long reasoning.

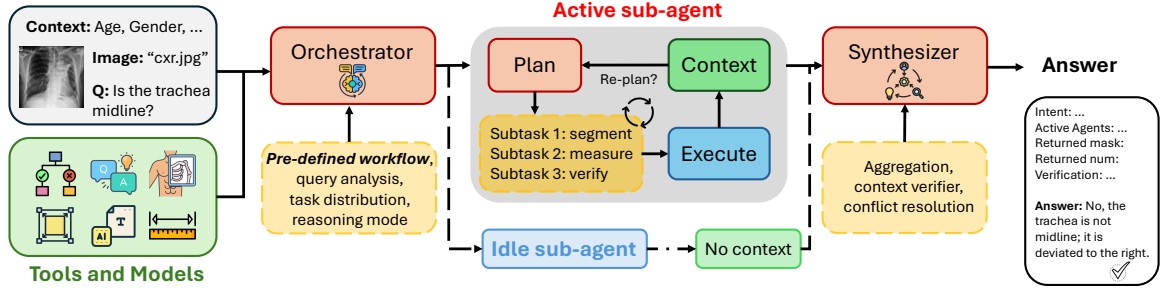

Figure 2: RadAgents framework. Each ABCDE subagent executes in parallel guided by clinical workflows, lowering latency, preserving isolation to avoid long-context drift, and improving trustworthiness.

## 2.1. Tool Set

To support radiologist-like reasoning, RadAgents integrates a comprehensive tool set that includes state-of-the-art machine learning models for specific tasks, general-purpose data processing utilities, and Python modules for measurement and calculation based on intermediate results, and for data utility.

- **ROI Segmentation.** Region-of-interest (ROI; e.g., anatomical structures and lesions) segmentation plays a central role in medical reasoning, as it provides interpretable visual evidence and often constitutes the first step in diagnostic workflows. For example, cardiothoracic ratio (CTR) calculation requires measuring both thoracic width and cardiac width from segmentation masks.
  **Tool list:** (a) CXAS, an anatomy segmentation model (Seibold et al., 2023), which can segment up to 157 anatomical structures relevant to chest radiography; (b) Biomed-Parser (Zhao et al., 2024a), a text-driven medical image parsing model that covers 82 major biomedical object ontologies, such as viral pneumonia.
- **Phrase Grounding.** Unlike object-level segmentation, phrase grounding aims to localize a finding described by free-text (e.g., "right lower lobe opacity") via a bounding box, providing finding-level evidence for generated outputs.
  **Tool list:** MAIRA-2 (Bannur et al., 2024), selected because it is trained on diverse (public and private) grounded datasets.
- **Measurement and Calculation.** Radiologists routinely assess the size, shape, and geometric relationships of ROIs to refine diagnoses, such as measuring the carinal angle or estimating pleural effusion volume for severity assessment.
  **Tool list:** We implement a suite of reusable Python scripts that perform geometric measurements and numeric calculations given either image inputs (e.g., segmentation masks, keypoints) or structured text. Further details are provided in Appendix B.
- **Visual Question Answering (VQA).** Medical VQA models enable the agentic system to handle flexible free-form queries, especially when it is unnecessary or overly costly to execute full visual reasoning pipelines.
  **Tool list:** We adopt MedGemma (Sellergren et al., 2025), which combines strong instruction-

following ability with medical knowledge. As a specialist CXR model, CheXagent is added as a complementary tool.

- **Report Generation.** Report generation models serve as references to initialize or supplement the final radiology report, particularly for routine findings.
  **Tool list:** We use the CheXpert Plus report generator (Chambon et al., 2024). MAIRA-2 is reused here to provide grounded visual evidence that can be integrated into the generated report.
- **Pathology Classification.** For certain pathology-specific pixel patterns, it is difficult to explicitly quantify the features needed for reasoning, and classification models become particularly valuable.
  **Tool list:** (a) A DenseNet-121 model from TorchXRayVision (Cohen et al., 2022), trained on four large-scale CXR datasets; (b) the VQA models, which can also be used in a classification mode by constraining their outputs.
- **Data Processing.** General data-processing utilities include a DICOM loader (with metadata parsing for fine-grained measurement), visualization tools, and basic preprocessing operations such as contrast adjustment and resizing, which standardize inputs for downstream tools.

### 2.2. Task-aware Subagents

Each subagent, also called the ABCDE agent, has a defined purpose and domain of expertise. Each is governed by a custom system prompt and maintains its own context window. The main scope and objectives of them are:

**Airway agent:** Systematically assess the central thorax for airway patency, alignment, and paratracheal lesions; for example, determine tracheal position (midline versus deviation).

**Breathing agent:** Survey the lungs and pleura for parenchymal and pleural pathology; for example, detect opacities (atelectasis, inltrate) and nodule.

**Circulation agent:** Evaluate the cardiac silhouette, mediastinum, and vessels; for example, compute the cardiothoracic ratio.

**Diaphragm agent:** Assess diaphragmatic integrity and look for subdiaphragmatic air; for example, compare right and left diaphragm height.

**Everything-else agent:** Identify chest wall (ribs and fractures), soft tissue, and foreign materials like medical devices.

Each subagent is equipped with an LLM and an individual **skill set** encoded in its system prompt. The skill set is a collection of reusable skill units, each defined as a reference tool chain together with decision thresholds for a specific clinical purpose. For example, computing the carinal angle requires first obtaining a segmentation mask of the tracheal bifurcation, then applying a geometric algorithm to identify the carina and main bronchi, and finally comparing the resulting angle against the normal range of 40–80 degrees.

The operational plan, i.e., which skills to invoke and in what order, is determined by the high-level intent distributed by the Orchestrator. Conditioned on this intent, each subagent performs step-by-step reasoning and tool use following the ReAct ("Reason + Act") paradigm (Yao et al., 2023); if a step fails or produces inconsistent evidence, the subagent triggers local re-planning to repair the failure. This behavior is agentic rather than a fixed automation script, making the system more robust to uncertainty while still

leveraging established clinical practice patterns. Details of the skill sets for each subagent are provided in Appendix B and A.

## 2.3. Global Controller Module

The global controller comprises the *Orchestrator* and the *Synthesizer*. The Orchestrator selects subagents and allocates tasks with appropriate patient context, and the Synthesizer integrates their outputs, verifies consistency, and resolves errors and conflicts. The major components are detailed below.

**Query analysis.** Given a query, the *Orchestrator* first analyzes the intent, extracts key clinical entities and objectives, and then drafts a high-level plan. It activates only the associated subagents (all other agents remain idle, incurring no additional computation cost) and sends them clear, goal-oriented instructions that specify the required clinical indicators without prescribing which tools to use. For example, for the query *"Is there lung opacity?"*, the Orchestrator generates a plan derived from our predefined **Workflow**, such as: *"Goal: (1) determine the existence of lung opacity; (2) if present, determine the type; (3) determine the location; and (4) verify the answer."*

This decoupling between the Orchestrator and the tool set greatly improves maintainability: otherwise, adding a new tool or updating an existing one would require rewriting the workflow. Instead, we introduce a skill layer as an intermediate abstraction and let each subagent, guided by language understanding, dynamically compose tools to accomplish a given skill. In this way, workflows, skills, and tools can be maintained and evolved independently. If a task is dispatched incorrectly, the receiving subagent raises a `SkillMismatchError` to request re-dispatch.

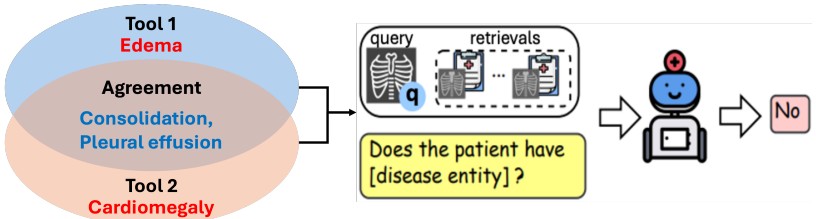

Figure 3: V-RAG Mechanism. When tool outputs conflict, the system retrieves top-$k$ clinically similar CXR studies as reference standards to verify the findings and resolve the disagreement (e.g., Edema and Cardiomegaly).

**Retrieval-augmented conflict resolution.** No tool is perfect, as its capabilities are bounded by model size and training data, and different tools may produce conflicting outputs. On the *Synthesizer* side, we therefore apply Visual Retrieval-Augmented Generation (V-RAG) (Chu et al., 2025): the Synthesizer retrieves clinically similar chest radiographs (using image embeddings from Rad-DINO (Perez-Garcia et al., 2025)) together with associated context such as patient notes, and leverages these exemplars to adjudicate discrepancies among tools (Figure 3 and Appendix D). This design mirrors routine radiologic practice, where clinicians consult prior cases and reference material to calibrate their interpretations.

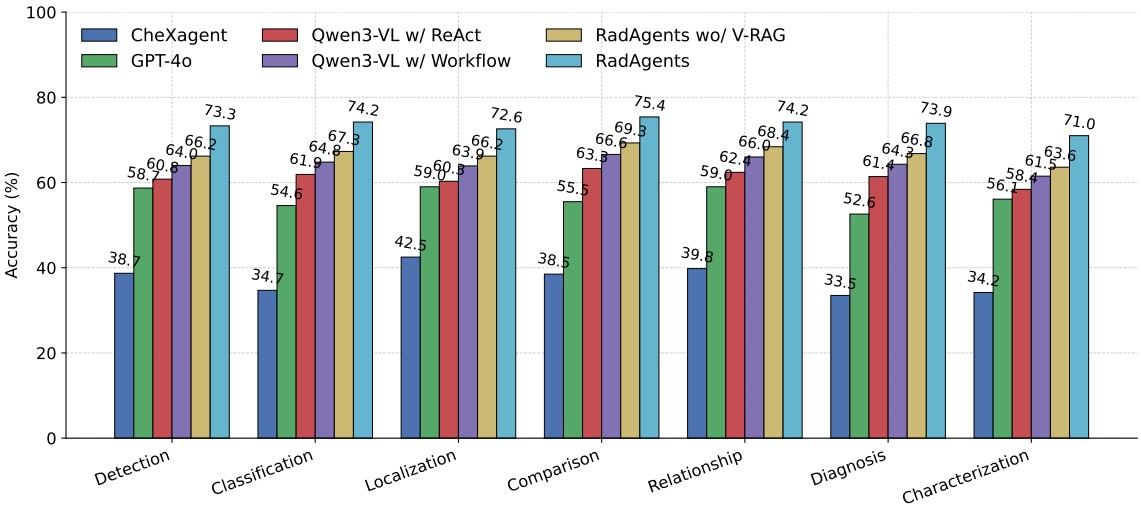

Figure 4: Performance on ChestAgentBench across different categories of questions.

**Short-term memory.** RadAgents maintains a shared short-term memory that caches patient-specific context, including demographics, clinical indications, acquisition information (e.g., AP vs. PA views), and metadata when DICOM images are provided. In addition, the short-term memory stores tool outputs, which are accessible to all agents. If an agent needs to call a tool whose result is already cached, it directly reads the cached output instead of re-invoking the tool. This mechanism prevents redundant computation, reduces latency, and improves consistency in multi-step analyses that repeatedly reference the same intermediate results.

## 3. Experiments

### 3.1. Experimental Setup

**Datasets.** To demonstrate the generality of **RadAgents**, we evaluate it on three benchmark datasets that closely mirror complex clinical workflows, offer sufficient task diversity, and are relatively easy to evaluate for correctness: (1) ChestAgentBench (Fallahpour et al., 2025) includes 2,500 questions derived from expert-validated clinical cases, covering seven core competencies and associated reasoning skills essential for CXR interpretation. (2) For the CheXbench (Chen et al., 2024) subset, following MedRAX, we focus on visual question answering (115 cases from Rad-Restruct (Pellegrini et al., 2023) and 123 cases from SLAKE (Liu et al., 2021)) and 380 fine-grained multimodal reasoning questions from OpenI [1]. (3) We further evaluate on the preprocessed multi-view and longitudinal MIMIC-CXR test set (2,231 cases) used in EditGRPO (Zhang et al., 2025).

**Baselines.** Unless otherwise specified, we instantiate all agents with Qwen3-VL-Instruct-8B. Additional results using GPT-4o as the agent core, with the same tool set as MedRAX, are provided in Appendix C. For comparison, we include: (1) specialist models CheXagent

---

1. https://openi.nlm.nih.gov/

Table 1: Accuracy (%) comparison on CheXbench.

| Model | VQA | | OpenI Reasoning | Overall |
|---|---|---|---|---|
| | Rad-Restruct | SLAKE | | |
| CheXagent | 57.1 | 78.1 | 59.0 | 64.7 |
| GPT-4o | 53.9 | 85.4 | 51.1 | 63.5 |
| Qwen3-VL w/ ReAct | 70.4 | 86.2 | 61.3 | 68.0 |
| Qwen3-VL w/ Workflow | 72.2 | 87.8 | 65.3 | 71.0 |
| RadAgents wo/ V-RAG | 71.3 | 87.8 | 66.3 | 71.5 |
| RadAgents | 76.5 | 89.4 | 69.2 | 74.6 |

and GPT-4o; (2) Qwen3-VL in a single-agent setting using ReAct (following MedRAX) without workflow steering; (3) Qwen3-VL (single agent) using ReAct guided by our workflow templates (including the full skill set). Unless otherwise noted, the number of retrieved exemplars for V-RAG is set to $k = 3$ (see Figure 7 and Appendix D for ablation study). We report results for two variants of RadAgents, with and without V-RAG.

**Metrics.** For ChestAgentBench and CheXbench, which consist of closed-ended questions, we report accuracy. For report generation, we use GREEN score (Ostmeier et al., 2024), which follows an LLM-as-a-judge paradigm and has been shown to align well with human judgments. Because the outputs are free-text sentences, this metric captures both clinical correctness and consistency.

### 3.2. Main Results

**ChestAgentBench.** Figure 4 compares the performance of different systems on the seven categories in ChestAgentBench. RadAgents achieves the best accuracy in every category, yielding an overall score of 73.6%, substantially higher than CheXagent (39.5%), GPT-4o (56.4%), Qwen3-VL w/ ReAct (61.3%), Qwen3-VL w/ Workflow (63.5%), and RadAgents without V-RAG (66.9%). The gains are consistent across tasks, with RadAgents outperforming the strongest non-agent baseline (Qwen3-VL w/ Workflow) by 7–10 points on most categories. The largest margins are observed on diagnosis (73.9% vs. 64.3%) and characterization (71.0% vs. 61.5%), which require synthesizing subtle imaging findings and clinical priors. Comparing RadAgents with and without V-RAG also reveals a clear benefit from visual retrieval: V-RAG contributes around 6–7 absolute points overall, suggesting that access to external image evidence is particularly helpful for fine-grained and high-level reasoning questions.

**CheXbench.** Table 1 reports results on CheXbench, which includes two VQA benchmarks (Rad-Restruct and SLAKE) and the OpenI image-text reasoning task. RadAgents again attains the highest overall accuracy (74.6%), improving upon both domain-specific CheXagent (64.7%) and the general-purpose GPT-4o (63.5%). On visual QA, RadAgents reaches 76.5% on Rad-Restruct and 89.4% on SLAKE, indicating strong capability in localized and fine-grained visual understanding. Qwen3-VL w/ Workflow and RadAgents w/o V-RAG are competitive, but RadAgents still provides a consistent 2–4 point advantage across all three sub-tasks. The OpenI reasoning task is more challenging for all models, yet RadA-

gents achieves 69.2% accuracy, outperforming RadAgents w/o V-RAG (66.3%) and other baselines. These results highlight that the proposed agentic workflow, together with visual retrieval, not only enhances structured VQA but also benefits more global image–text reasoning.

**MIMIC-CXR Report Generation.** Figure 5 reports GREEN scores on the MIMIC-CXR test set under the multi-view and longitudinal setting, which better reflects real clinical workflows. RadAgents attains the highest GREEN score of 51.4, outperforming RadAgents w/o V-RAG (46.1), GPT-4o w/ ReAct (42.3), GPT-4o w/ Workflow (41.7), plain GPT-4o (34.2), and CheXagent (23.6). The relatively marginal improvement of the workflow-based variants over plain GPT-4o suggests that simply feeding long multi-study contexts to a single agent is insufficient, as the model can become "lost in the middle" (Liu et al., 2024) and under-utilize information dispersed across the sequence. The sizeable gap between RadAgents and its ablated variant highlights that visual retrieval and our specialized multi-agent design are particularly beneficial for generating temporally consistent reports conditioned on multiple studies.

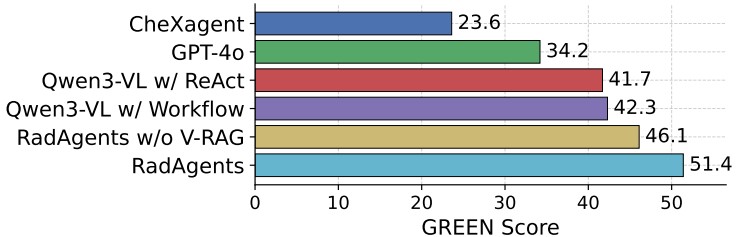

Figure 5: Multi-view and longitudinal performance on MIMIC-CXR test set.

### 3.3. Ablation Study

We investigate how the capability of the core LLM (i.e., model scale/parameters) influences RadAgents by varying the backbone Qwen3-VL-Instruct model from 4B to 8B and 30B. Since RadAgents contains multiple roles, in each ablation we upgrade only a single component, either the Orchestrator or the Synthesizer, while keeping all other agents at the default 8B model[2]. For example, when evaluating the Synthesizer, the Orchestrator and all subagents use the 8B model, whereas the Synthesizer uses the 30B model. This setup allows us to separately assess (a) whether a stronger Orchestrator can better orchestrate subagents (i.e., activate the correct specialists and deliver precise workflows), and (b) whether a stronger Synthesizer can more effectively resolve conflicts between tools and subagents.

The evaluation set comprises 100 representative cases: (a) 50 VQA instances randomly sampled from the MS-CXR test set, querying the existence and attributes of abnormalities (e.g., size and severity), and (b) 50 report-generation cases from the MIMIC-CXR test set, covering medium to high complexity. Details of the underlying datasets are provided in Appendix C.

---

2. Running all five agents with 30B models in parallel would be prohibitively expensive.

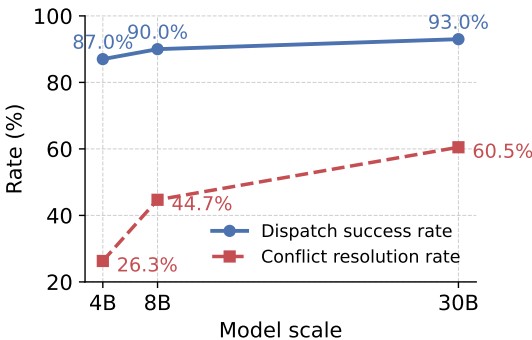

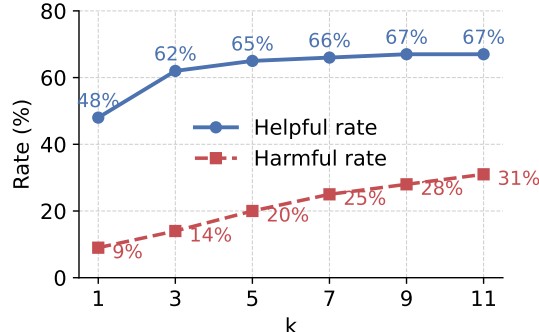

Figure 6: Effect of model scale on Orchestrator dispatch success and Synthesizer conflict-resolution rates.

Figure 7: Pilot study for $k$ selection in V-RAG, showing that larger $k$ increases the helpful retrieval rate but also raises the harmful rate.

Figure 6 shows that the dispatch success rate, defined as the proportion of cases where the correct subagents are activated and no "request re-dispatch" error is raised, increases from 87.0% to 93.0% as the Orchestrator model size grows from 4B to 30B. This indicates that stronger language understanding helps distribute tasks more reliably, though the gain is relatively modest because our hybrid search design already resolves most dispatch ambiguities. In contrast, the conflict-resolution rate of the Synthesizer (measured over the 38 cases with inter-tool disagreements, 9 from VQA and the remainder from report generation) improves substantially, rising from 26.3% (4B) to 44.7% (8B) and 60.5% (30B). These results suggest that conflict resolution is considerably more sensitive to model capacity than task dispatch, and that investing capacity in the Synthesizer is crucial for reliably reconciling heterogeneous tool and agent outputs.

## 4. Discussion

RadAgents leverages radiologist-inspired workflows, tool-augmented reasoning, and visual retrieval to achieve robust CXR interpretation across three complex benchmarks. Despite these advancements, the current framework has limitations. First, performance is intrinsically upper-bounded by the capabilities of the underlying tools; specifically, the current reliance on tools optimized for frontal-view X-rays limits robustness on lateral views. Second, the interaction between the orchestrator and tools is unidirectional: while the synthesizer can resolve conflicts via evidence weighting, it cannot iteratively guide or correct upstream tool outputs (e.g., refining an imperfect segmentation mask). Additionally, the multi-agent architecture incurs higher computational costs compared to end-to-end baselines. Future work will address these challenges by optimizing efficiency through dynamic agent selection and extending the framework to 3D modalities (e.g., CT and MRI) via modality-specific sub-agents, alongside prospective studies to validate clinical impact.

## Acknowledgments

We would like to thank other members of Oracle Health AI for their support while developing our system, and Raefer Gabriel, Sri Gadde, Mark Johnson, Devashish Khatwani, Yuan-Fang Li, Anit Sahu, Praphul Singh, and Vishal Vishnoi for insightful feedback and discussions.

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

## Appendix A. Pathology-aware Skill Set (Examples)

- **Lines, Tubes, and Devices.**
  *Goal:* Identify device type, localize position, and verify termination points (e.g., ET tube depth).
  *Workflow:* (1) **Detection:** Use BiomedParser with prompts (e.g., "Endotracheal tube", "CVC") to generate device masks. (2) **Anatomical Localization:** Overlay device masks with anatomical masks (Trachea, Carina, SVC) from CXAS. (3) **Verification:** Use Python scripts to measure distances (e.g., ET tip to Carina) or check containment (e.g., NG tube tip inside Stomach mask). Malposition is flagged based on geometric thresholds.

- **Pneumothorax.**
  *Goal:* Confirm presence, laterality, and estimate size.
  *Workflow:* (1) **Screening:** Use DenseNet-121 to filter negative cases based on probability thresholds. (2) **Localization:** Deploy MAIRA-2 with prompts like "Pneumothorax" or "Pleural line" to generate bounding boxes. (3) **Side Determination:** Map bounding box centroids to Left/Right Lung masks from CXAS. (4) **Characterization:** Query MedGemma (VQA) with cropped regions to estimate size (small/-moderate/large) and check for tension physiology.

- **Pleural Effusion.**
  *Goal:* Detect presence, laterality, and quantify volume.
  *Workflow:* (1) **Segmentation:** Use CXAS to segment Lung fields and Costophrenic Angles. (2) **Detection:** Use BiomedParser to segment "Fluid" or "Effusion" regions. (3) **Geometric Measurement:** Calculate the vertical height of the fluid mask relative to the total lung height. (4) **Quantification:** Apply rule-based thresholds (e.g., $< 1/4$ lung height = Small; $> 1/2$ = Large) and check for costophrenic angle blunting.

- **Lung Opacity (Pneumonia, Mass, Atelectasis, etc.).**
  *Goal:* Differentiate pathology type (texture) and precise anatomical location.
  *Workflow:* (1) **Classification:** Use DenseNet-121 to obtain probability distributions for various opacity types. (2) **Localization:** Use MAIRA-2 to generate bounding boxes for high-probability findings. (3) **Anatomy Mapping:** Overlay bounding boxes with CXAS lung lobe masks to assign location (e.g., RUL, LLL). (4) **Texture Analysis:** Use MedGemma (VQA) to describe pixel patterns (e.g., "patchy/fluffy" for pneumonia vs. "linear/plate-like" for atelectasis) to confirm diagnosis.

- **Cardiac Silhouette.**
  *Goal:* Assess heart size and calculate Cardiothoracic Ratio (CTR).
  *Workflow:* (1) **Segmentation:** Use CXAS to segment the Heart and Thoracic Cavity/Lungs. (2) **Calculation:** Execute Python scripts to measure the maximum horizontal cardiac width and thoracic width. (3) **Thresholding:** Calculate CTR (Cardiac Width / Thoracic Width); if $> 0.5$, classify as Cardiomegaly/Enlarged.

- **Mediastinum and Hilar Regions.**
  *Goal:* Evaluate mediastinal width/contours and hilar enlargement.

*Workflow:* (1) **Segmentation:** Use CXAS to segment the Mediastinum and Aortic Knob. (2) **Measurement:** Measure superior mediastinal width via Python script (threshold > 8 cm for widening). (3) **Hilar Assessment:** Use MAIRA-2 to detect "Hilar enlargement" or "Mass"; if detected, use VQA to characterize potential lymphadenopathy or vascular prominence.

- **Skeletal Abnormalities.**
  *Goal:* Detect and localize fractures or bone lesions.
  *Workflow:* (1) **Detection:** Use BiomedParser to parse "Fracture" or "Bone lesion" into masks. (2) **Localization:** Intersect finding masks with CXAS skeletal masks (Ribs, Clavicles, Spine) to identify specific bones (e.g., "Right 6th Rib"). (3) **Confirmation:** Use VQA on the ROI to verify cortical disruption and rule out false positives from vascular overlap.

## Appendix B. Anatomy-aware Skill Set (Examples)

- **Cardiomegaly (Cardiothoracic Ratio, CTR).** Given heart and bilateral lung masks, we first identify the axial row where the heart mask attains its maximal width. On this row, we extract the leftmost and rightmost pixels of the heart mask to obtain the cardiac width, and the leftmost and rightmost pixels of the union of the left- and right-lung masks to obtain the thoracic width. The cardiothoracic ratio is then

$$\text{CTR} = \frac{\text{cardiac width}}{\text{thoracic width}}.$$

  Comparing CTR to a view-specific clinical threshold (e.g., CTR > 0.5 on PA view) yields a binary cardiomegaly label.

- **Carina Angle.** From the trachea bifurcation mask we compute a concave hull and locate the lowest point where the trachea splits, defining the carina point. At 10%, 20%, and 30% of the mask height below the carina, we find the leftmost and rightmost side points and average them to estimate the centerlines of the left and right main bronchi. The carina angle is defined as the interior angle between the two vectors from the carina point to the left and right bronchus centerline points, respectively. This angle is compared against a normal reference range to determine whether the carina angle is abnormal.

- **Tracheal Deviation.** Using trachea, trachea bifurcation, and vertebrae (T1–T7) masks, we first suppress the carina region within the trachea mask, retaining the upper trachea. From the vertebral masks, we extract posterior spinous-process points that lie within the vertical extent of the trachea mask. For each such vertebral level, we compare the horizontal position of the trachea to the corresponding spinous-process point and assign a local label (left, right, centered). The final tracheal deviation label is obtained via majority voting across levels: predominant left labels indicate left deviation, predominant right labels indicate right deviation, and balanced or centered labels indicate no deviation.

- **Mediastinal Widening.** From upper mediastinum and bilateral lung masks, we find the row where the mediastinum mask is widest. On this row, we extract the leftmost and rightmost mediastinum pixels to obtain the mediastinum width and the leftmost and rightmost pixels of the combined lung masks to obtain the thoracic width at the same level. The mediastinum-to-thoracic width ratio

$$\text{Ratio} = \frac{\text{mediastinum width}}{\text{thoracic width}}$$

is compared against an expert-defined threshold to determine the presence of mediastinal widening.

- **Aortic Knob Enlargement.** Using aortic arch, descending aorta, trachea, and trachea bifurcation masks, we first localize the aortic knob with the aortic arch mask. The trachea mask, restricted to the vertical range of the aortic arch and excluding the bifurcation, provides the starting point of the knob measurement: *Point A* is the average $x$-coordinate of the left tracheal wall in this range. From the upper 30% of the descending aorta mask, we extract the innermost $x$-coordinate (*Point B*) and the outermost $x$-coordinate on the same row (*Point C*). *Point D* is the outermost $x$-coordinate of the aortic arch mask. The aortic knob width is defined as the horizontal distance from Point A to the farthest of Point C or Point D. The tracheal width is computed as the median horizontal width of the trachea mask (excluding the bifurcation). The diagnostic index is

$$\text{Ratio} = \frac{\text{aortic knob width}}{\text{tracheal width}},$$

and aortic knob enlargement is declared when this ratio exceeds an expert-defined threshold.

- **Ascending Aorta Enlargement.** With ascending aorta, heart, trachea, and trachea bifurcation masks, we first identify the most right-sided point of the heart mask (*Point A*) and the most right-sided point of the trachea mask excluding the bifurcation (*Point B*), which approximates the inner boundary of the right lung. We construct a reference line connecting Points A and B and then isolate the portion of the ascending aorta mask that extends beyond this line toward the right. Let $A_{\text{beyond}}$ be the area of the ascending aorta beyond the reference line and $A_{\text{total}}$ the total ascending aorta area. The diagnostic index is

$$\text{Ratio} = \frac{A_{\text{beyond}}}{A_{\text{total}}},$$

and if this ratio exceeds an expert-defined threshold, the ascending aorta is labeled enlarged.

- **Descending Aorta Enlargement.** From the descending aorta, trachea, and trachea bifurcation masks (with the heart mask used to define the thoracic segment), we retain only the descending aorta portion lying above the inferior border of the heart, corresponding to the thoracic aorta. Within this region, we identify the row where the aortic width is maximal and extract the leftmost and rightmost $x$-coordinates at this

row, yielding the maximum thoracic descending aorta width. From the trachea mask, we compute the median tracheal width across rows and record the corresponding left/right $x$-coordinates at the median row. The diagnostic index is

$$\text{Ratio} = \frac{\text{descending aorta width}}{\text{tracheal width}},$$

and descending aorta enlargement is indicated when this ratio exceeds an expert-defined threshold.

## Appendix C. Vanilla-MedRAX-comparable experiments using GPT-4o

### C.1. Implementation Details of RadAgents

RadAgents is implemented on top of the LangGraph agentic framework, and is conceptually inspired by MedRAX (Fallahpour et al., 2025). The core engine of each agent can in principle be any LLM; in this work we focus on open-source, lightweight models, specifically Qwen3-VL-Instruct series, which exhibit strong instruction-following and reasoning capabilities and supports long-context input that is crucial for our agentic system. Because we explicitly encode clear, radiologist-style workflows into the agents, even the smaller models can reliably follow the prescribed procedures in most cases. To ensure transparency and enable debugging and analysis, we log the full execution trajectory and all intermediate outputs for each run.

For workflow extraction, we adopt hybrid retrieval over the textual workflow descriptions, combining BM25 with dense similarity based on the Snowflake-arctic-embed-m-v1.5 embeddings. Skills and tools are exposed as structured JSON APIs, and agents issue calls by constructing precise JSON objects that specify the target tool together with all required arguments (e.g., image paths and text prompts).

### C.2. Experimental Setup

Here we use GPT-4o as the core LLM engine instead of Qwen3-VL, and, to make our system comparable to vanilla MedRAX, we adopt the same tool set. Considering the cost of API calls, we perform our evaluation on smaller-scale datasets that are less structurally complex than ChestAgentBench but exhibit increasing reasoning complexity: 181 MS-CXR (Boecking et al., 2022) VQA cases for existence and attribute (E&A) queries about abnormalities, 785 VQA cases for comparison and progression (C&P) of abnormalities, and 181 MIMIC-CXR two-view frontal report-generation cases.

### C.3. Results

**Existence and attributes.** The VQA questions cover seven common findings in CXR: atelectasis, cardiomegaly, consolidation, edema, lung opacity, pleural effusion, and pneumothorax, derived from the standard test split of MS–CXR. Each image receives the following prompt:

```
<image> Describe if [finding] is present; if present, describe [attributes].
```

**Comparison and progression.** We use MS–CXR–T to assess stability, improvement, or worsening of a specific positive finding (consolidation, edema, pleural effusion, or pneumothorax). We only retain cases where the metadata indicates a **consensus** among human reviewers. We pose a comparative question that explicitly references the prior study. The prompt template is:

```
Given current image <image>, and previous image <image>, decide if [finding] is
improving, stable, or worsening.
```

**Report generation.** We construct a MIMIC–CXR subset aligned with MS–CXR identities so that findings queried in VQA are represented in the corresponding reports. Prompts request generation of the *Findings* section, and all agents are activated by default.

Table 2 summarizes performance across the three evaluation settings. RadAgents with V-RAG achieves the best results on all tasks. Relative to the ablated agent without V-RAG, it improves GREEN on VQA (E&A) from 0.5841 to 0.6032 (+0.0191, ∼3.3%), raises VQA (C&P) accuracy from 48.0% to 50.9% (+2.9 points, ∼6.0%), and boosts report-generation GREEN from 0.3821 to 0.4527 (+0.0706, ∼18.5%). Augmenting GPT-4o with ReAct and our workflow also yields consistent gains over the base LLM; for example, GREEN on VQA (E&A) increases from 0.2127 (GPT-4o) to 0.4619 (GPT-4o+ReAct) and 0.5351 (GPT-4o+Workflow), while VQA (C&P) accuracy rises from 18.2% to 41.9% and 45.5%, respectively. Nonetheless, even the strongest GPT-4o+Workflow baseline remains substantially behind RadAgents, trailing by 0.0681 GREEN on VQA (E&A), 5.4 accuracy points on VQA (C&P), and 0.0686 GREEN on report generation. Overall, the ranking of methods is largely consistent across tasks, suggesting that the benefits of structured tool use and V-RAG transfer from explanation-style VQA to longitudinal comparison questions and free-text report generation.

Table 2: Performance comparison between different setting when using gpt-4o as the core LLM engine in RadAgents.

| Method | VQA (E&A) (GREEN) | VQA (C&P) (Acc. %) | Report Generation (GREEN) |
|---|---|---|---|
| CheXagent | 34.3 | 34.1 | 18.3 |
| GPT-4o | 21.3 | 18.2 | 31.4 |
| GPT-4o w/ ReAct | 46.2 | 41.9 | 33.3 |
| GPT-4o w/ Workflow | 53.5 | 45.5 | 38.4 |
| RadAgents w/o V-RAG | 58.4 | 48.0 | 38.2 |
| RadAgents | **60.3** | **50.9** | **45.3** |

# Appendix D. Details of V-RAG

**Multimodal retrieval.** We retrieve images and their textual descriptions that align with the features of target medical images following (Chu et al., 2025). These references, rich

in visual and textual medical details, guide response generation. To obtain embeddings, we use Rad-DINO, which provides robust representations across diverse CXR image types. For each image $X_{img}$, we extract its embedding $E_{img} = \mathbf{R}^d$, with $d = 768$, and store them in the embedding memory $\mathcal{M}$.

**Corpus.** The retrieval index consists exclusively of the MIMIC-CXR official training split. We strictly enforce patient-level exclusion (via Subject ID), ensuring test patients are completely disjoint from the retrieval corpus. For external benchmarks (e.g., ChestAgentBench), the data originates from distinct sources, ensuring zero overlap with the MIMIC-CXR index.

**Sensitivity of $k$.** We study how retrieval quality changes with the number of retrieved studies $k$ with the sampled 100 cases from the MS-CXR test set used in this study. For each setting, we compute two metrics: *helpful-rate*, the percentage of retrieved studies that improve the answer, and *harmful-rate*, the percentage that hurt the answer. As shown in Figure 7, As k increases, the harmful rate grows more quickly than the helpful, e.g., the helpful-rate increases from 0.48 at $k = 1$ to 0.65 at $k = 5$, while the harmful-rate also rises from 0.09 to 0.20. We hypothesize this is due to longer contexts imposing a heavier reasoning burden and increasing hallucination, consistent with prior LLM findings. This illustrates a trade-off: retrieving more studies provides greater chances of including helpful evidence but also increases the risk of introducing misleading content. To balance these effects, we choose $k = 3$ by default, achieving a helpful-rate of 0.62 with a moderate harmful-rate of 0.14.

**Augmented Inference.** In the inference stage, we encode the query image $X_q$ to obtain its embedding. We then retrieve the top-$k$ most similar images from $\mathcal{M}$, represented as $(I_1, \ldots, I_k)$ with their corresponding reports $(R_1, \ldots, R_k)$. These references are appended to the input of multimodal LLM to guide generation. The prompt is structured as:

```
This is the i-th similar image and its report for your reference.  [Reference]_i
...  According to the query image and the references, [Question] [Query Image].
```

where each reference is denoted as $(I_i, R_i)$.

For efficient retrieval during inference, we build $\mathcal{M}$ using FAISS [3], a GPU-accelerated vector search system. We employ approximate kNN with the Hierarchical Navigable Small World (HNSW) algorithm (Malkov and Yashunin, 2018), enabling retrieval of the top-$k$ most similar images in $\mathcal{M}$.

## Appendix E. Additional Results and Analysis

### E.1. Entity-level Evaluation for Report Generation

Evaluating the quality of generated radiology reports is non-trivial. Early works adopted general-domain natural language processing metrics such as ROUGE (Lin, 2004) and BLEU (Papineni et al., 2002). While these metrics are widely used for text evaluation, they treat

---

3. https://github.com/facebookresearch/faiss

differences in wording the same as clinically significant errors, failing to reflect medical accuracy. To address this limitation, clinically informed evaluation metrics, such as CheXbert (Smit et al., 2020), RadGraph (Jain et al., 2021), GREEN (Ostmeier et al., 2024), and RaTEScore (Zhao et al., 2024b), have been proposed to better assess clinical correctness and utility. CheXbert is based on multi-label classification results for 5 or 13 diseases (along with one extra "normal" label). RadGraph considers literal entity agreement considering the positive or negative context of each entity. GREEN judges recall and precision errors by LLM prompting. RaTEScore is inspired by RadGraph but less sensitive to phrasing by an F1-like computation which allows semantic matching between entities based on a cosine similarity. The metrics are computed using their official and standardized implementations: RADGRAPH-F1[4], CHEXBERT-F1[5], RATE SCORE[6], and GREEN[7].

Table 3: Performance Comparison on Report Generation Metrics. The proposed RadAgents achieves the highest scores across all standard metrics.

| Settings / Metrics | CheXbert-macro-F1 (14) | RadGraph-F1 | RaTE |
|---|---|---|---|
| CheXagent | 29.2 | 12.7 | 44.4 |
| GPT-4o | 22.4 | 15.3 | 49.8 |
| Qwen3-VL w/ ReAct | 44.9 | 17.1 | 53.6 |
| Qwen3-VL w/ Workflow | 45.5 | 17.2 | 53.9 |
| RadAgents w/o V-RAG | 49.1 | 18.1 | 55.8 |
| **RadAgents (Ours)** | **53.2** | **19.4** | **58.5** |

## E.2. Computational Cost and Latency Analysis

To assess the deployment feasibility of RadAgents, we conducted a detailed analysis of inference latency and computational cost.

**Experimental Setup and Mechanism.** A critical efficiency feature of RadAgents is its sparse activation design. As detailed in Section 2.3, the Orchestrator activates strictly those sub-agents required for a specific query (e.g., a query regarding "bone fractures" triggers only the relevant specialist, leaving the "Lung Opacity" agent idle). Consequently, the system rarely incurs the computational overhead of running all 7 agents simultaneously; idle agents consume zero active compute resources.

Performance was measured on an on-premise node equipped with $8\times$ NVIDIA RTX A5000 (24GB) GPUs. To reflect a practical deployment environment, we utilized quantized models served via Ollama backends with LangGraph orchestration. We report two key metrics: (1) **Latency**: End-to-end wall-clock time per case (seconds), and (2) **Compute Cost**: Cumulative GPU usage per case (GPU-seconds).

---

4. https://pypi.org/project/radgraph/0.1.2/

5. https://pypi.org/project/f1chexbert/

6. https://pypi.org/project/RaTEScore/0.5.0/

7. https://pypi.org/project/green-score/0.0.8/

**Quantitative Results.** We compared the proposed RadAgents (in both Sequential and Parallel execution modes) against single-agent baselines across three benchmarks. The results are summarized in Table 4.

Table 4: Latency and Cost Analysis across benchmarks. Metrics are reported as **Wall-Clock Time (sec) / Compute Cost (GPU-sec)**. The Parallel configuration significantly reduces latency compared to sequential execution.

| Configuration | CheXbench (sec / GPU-sec) | ChestAgentBench (sec / GPU-sec) | Report Gen (sec / GPU-sec) |
|---|---|---|---|
| Single-Agent ReAct | 16 / 30 | 25 / 50 | 50 / 110 |
| Single-Agent Workflow | 18 / 36 | 29 / 64 | 58 / 140 |
| RadAgents (Sequential) | 38 / 89 | 62 / 178 | 120 / 368 |
| **RadAgents (Parallel)** | **26 / 83** | **41 / 164** | **85 / 340** |

**Analysis. Parallelism Efficiency:** Running sub-agents in parallel reduces wall-clock latency by approximately 30–40% compared to the sequential variant. For complex queries in ChestAgentBench, the system achieves an average latency of 41 seconds, rendering it responsive enough for asynchronous clinical workflows.

**Cost-Benefit Trade-off:** While the multi-agent architecture incurs roughly 2–3× the compute cost of a simple Single-Agent baseline, this is a necessary trade-off to facilitate the complex reasoning that drives the observed 10%+ performance gains. In the context of high-stakes medical diagnosis, where accuracy and traceability are paramount, this increased computational investment is justified.

### E.3. Systematic Error Analysis

To evaluate the robustness of the Orchestrator/Sub-agents and the stability of the agentic workflows, we analyzed the frequency of workflow deviations across three benchmarks. Specifically, we tracked two key metrics:

1. **Workflow-free (ReAct) Fallback Rate**: The percentage of queries where the Orchestrator could not match a pre-defined clinical workflow and defaulted to a generalist ReAct loop.

2. **SkillMismatchError / Re-dispatch Rate**: The frequency with which an assigned sub-agent rejected a task (due to scope mismatch) or failed, necessitating a re-dispatch by the Orchestrator.

**Quantitative Results.** The average error frequencies are summarized in Table 5.

**Interpretation.** The variance in error rates reflects the distinct nature of the evaluation datasets:

Table 5: Systematic Error Frequency Analysis. **ChestAgentBench** exhibits higher fall-back rates due to its open-ended query nature, whereas **MIMIC-CXR** report generation follows a highly structured routine, resulting in minimal deviations.

| Datasets | Workflow-free (ReAct) Fallback Rate | SkillMismatchError / Re-dispatch Rate |
|---|---|---|
| CheXbench | 10.4% | 5.3% |
| ChestAgentBench | **18.1%** | **9.3%** |
| MIMIC-CXR Report Gen | 2.1% | 4.0% |

- **ChestAgentBench** contains a higher proportion of open-ended, mixed-intent queries. This complexity forces the Orchestrator to utilize the fallback ReAct mechanism more frequently (18.1%) and results in higher rates of internal re-dispatching (9.3%) to resolve ambiguities.

- **MIMIC-CXR Report Generation** typically follows a fixed, routine clinical workflow (e.g., standard frontal/lateral review). Consequently, it exhibits the highest stability with a minimal workflow-free fallback rate (2.1%).

- **CheXbench** falls between these two extremes, representing a balanced mix of structured classification tasks and moderately complex reasoning queries.

## Appendix F. Demonstration

An end-to-end execution trace for the query "Is the trachea midline?". (a) Dispatch: The Orchestrator routes the query to the Airway Agent based on intent analysis. (b) Tool Execution: The sub-agent invokes the segmentation tool; note the incorrect "midline" classification in the raw tool metadata despite successful segmentation. (c) Conflict Detection: The Synthesizer performs a cross-modal check using a VQA model, detecting a discrepancy between the tool's heuristic and the visual assessment. (d) Resolution: The Synthesizer triggers V-RAG to retrieve Top-3 similar cases. By synthesizing the retrieved evidence with its own reasoning, the system corrects the tool error to output the final accurate diagnosis of "deviated to the right."

## Execution Trace: Query "Is the trachea midline?"

### ⊟ STEP 1: ORCHESTRATION & DISPATCH

```
[ORCHESTRATOR] Analyzing Query Context...
Query: ``Is the trachea midline?''    Intent: Anatomic Localization
(Airway)
Strategy: Activate [Airway_Agent]   Dispatch: > Executing Airway_Agent...
```

### ⚒ STEP 2: SUB-AGENT REASONING & TOOL EXECUTION

```
[AIRWAY_AGENT] Action: Calling tool
`check_tracheal_position`
Params: { ``organ'': ``Trachea'' }
[TOOL_OUTPUT] Status: SUCCESS
Metrics: { ``area_pixels'': 12490, ``centroid'':
[157.8, 470.0],
``heuristic_classification'': ``midline'' <--
(Incorrect) }
[AIRWAY_AGENT] Tool execution complete. >
Handoff to Synthesizer.
```

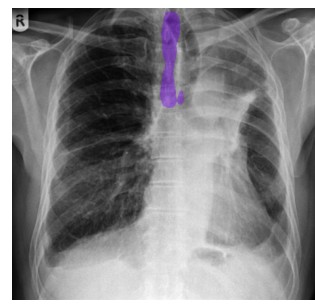

*Tool Segmentation Visualization*

### ⚠ STEP 3: CONFLICT DETECTION (SYNTHESIZER)

```
[SYNTHESIZER] Phase 1: Verification
> Tool Claim: ``Trachea is midline'' (based on centroid heuristic).
> Action: Cross-check with VQA (Global View).   > VQA: ``The trachea
appears deviated to the right.''
```

> ! ALERT: Tool output contradicts visual assessment.
> ! Status: UNCERTAIN. Triggering V-RAG Resolution...

### ✅ STEP 4: RESOLUTION (V-RAG + SYNTHESIS)

```
[SYNTHESIZER] Phase 2: Resolution via V-RAG
> Action: Retrieving Top-3 clinically similar cases...   [RETRIEVAL] Found
3 references (Sim > 0.85):
```

- 1. Case_782: ``Trachea deviated to right'' (Visual match: High)
- 2. Case_104: ``Rightward deviation due to effusion'' (Visual match: High)
- 3. Case_993: ``Midline trachea'' (Visual match: Low)

```
[FINAL REASONING]
``Although segmentation suggests midline, VQA and 2/3 retrieved cases
confirm rightward deviation. The tool's centroid heuristic is likely misled
by patient rotation.''
```

> [FINAL ANSWER] ``No, the trachea is not midline; it is deviated to the right.''
> Confidence: High (Supported by V-RAG Evidence)

