# OpenReview forum: "RadAgents: Multimodal Agentic Reasoning for Chest X-ray Interpretation with Radiologist-like Workflows"
_MIDL.io/2026/Conference — MIDL 2026 Poster_

### Official Review · Reviewer_GQVi · 2026-01-07

**Confidence:** 4
**Preliminary Rating:** 2
**Final Rating:** 3

**Summary:**

This paper proposes RadAgents, a multi-agent, tool-augmented framework for chest X-ray (CXR) interpretation that aims to mimic radiologist-like workflows. A key technical component is Visual Retrieval-Augmented Generation (V-RAG), where the Synthesizer retrieves visually similar CXRs (via Rad-DINO embeddings) to adjudicate conflicting tool outputs. Experiments are conducted on ChestAgentBench, CheXbench, and MIMIC-CXR multi-view/longitudinal report generation.

**Strengths:**

-	Clear alignment with radiologist-style workflows
-	Broad empirical evaluation across multiple CXR reasoning benchmarks
-	Demonstrates that open-source mid-sized VLMs can be competitive when embedded in structured pipelines.

**Weaknesses:**

-	Missing key baselines. RadFabric and MedRAX are the most closely related prior works, both proposing agentic, tool-augmented systems for chest X-ray interpretation. Although they are discussed in the related work, neither is included as a direct baseline in the main experiments. How does RadAgents compare to them in terms of accuracy, robustness, and computational cost?
-	Limited novelty: the method is heavily built upon MedRAX, adding engineering components to improve performance.
-	Will V-RAG introduce harmful retrievals?
-	No reporting of inference latency or computational cost, despite system complexity.

**Detailed Comments:**

See above.

**Justification Of Final Rating:**

The rebuttal clarifies implementation details and improves presentation, which partially addresses earlier concerns. While the error handling relies on post hoc synthesis rather than true agent-level replanning, the system demonstrates a reasonably engineered integration of multiple tools with consistent empirical gains.

**Justification Of The Preliminary Rating:**

While the paper is carefully engineered and empirically strong, it primarily represents an incremental system-level refinement built on predefined clinical workflows and MedRAX. The lack of direct comparison to RadFabric and MedRAX significantly undermines the novelty and empirical positioning.

**Questions To Address In The Rebuttal:**

See above

---

> ### Author Response · Authors · 2026-01-24
>
> Thank you for taking the time to review our paper and for providing such insightful comments. If our response addresses your concerns, we would greatly appreciate it if you could consider improving your review score.
>
>
> **(1) Missing key baselines. RadFabric and MedRAX are the most closely related prior works, both proposing agentic, tool-augmented systems for chest X-ray interpretation.**
>
> Thank you for the comment. Below we clarify how our experiments relate to them and report the corresponding accuracy comparisons available under our evaluation protocol.
>
> **1. MedRAX** is a ReAct-style single-agent tool-orchestration framework for CXR interpretation. In our main experiments, the baseline “Qwen3-VL w/ ReAct” instantiates the same ReAct controller-and-tools execution loop under our unified tool API (i.e., the same baseline paradigm as MedRAX), enabling a direct accuracy comparison in our setting.
>
> Concretely, RadAgents outperforms the ReAct baseline across all three benchmarks reported in the paper:
>
> * **ChestAgentBench overall:** 73.6% (RadAgents) vs 61.3% (Qwen3-VL w/ ReAct)
> * **CheXbench overall accuracy:** 74.6% (RadAgents) vs 68.0% (Qwen3-VL w/ ReAct)
> * **MIMIC-CXR report generation (GREEN):** 51.4 (RadAgents) vs 42.3 (ReAct-style baseline reported in Fig. 5)
>
> **2. RadFabric** is a closed-sourced agentic radiology system, and it does not report results on our three benchmarks under the same evaluation protocol and does not specify the used evaluation datasets; therefore, we cannot provide a direct, apples-to-apples accuracy number for RadFabric in our main experimental tables.
>
> **(2) Limited novelty: the method is heavily built upon MedRAX, adding engineering components to improve performance.**
>
> We respectfully disagree that RadAgents is merely an engineering extension of MedRAX. While both works aim to utilize tools for CXR analysis, RadAgents represents a fundamental paradigm shift in cognitive architecture, moving from "sequential tool use" to "collaborative visual reasoning." The specific novelties are three-fold:
>
> **1. Architectural Shift: From Sequential ReAct to Hierarchical Collaboration**
>
> * **MedRAX (Prior Art):** Relies on a single-agent ReAct loop. This is a linear, "single-thread" process where one mistake in the reasoning chain leads to irreversible error propagation.
> * **RadAgents (Ours):** Introduces a **Orchestrator-Worker-Synthesizer** architecture. This decouples "planning" from "execution," enabling parallel tool invocation and distinct role specialization. This is not an engineering tweak but a different topological design that solves the "forgetting" and "context overflow" issues inherent in single-agent loops.
>
> **2. Reasoning Shift: From Text-Only Handoff to Visual Synthesis**
>
> * **MedRAX:** Treats tools as black boxes that output text (e.g., "opacity in left lung"). The agent never "sees" the segmentation mask, leading to hallucination if the text description is vague.
> * **RadAgents:** Performs **Interleaved Visual Reasoning**. The Synthesizer actively consumes visual outputs (segmentation masks, bounding boxes) overlaid on the original image. This transforms the system from a "text-processing agent" to a "visually-grounded agent," allowing it to verify spatial relationships (e.g., "is the catheter tip strictly inside the SVC?") that MedRAX fundamentally cannot handle.
>
> **3. Robustness Mechanism: From Blind Execution to Active Verification**
>
> * **MedRAX:** Lacks a conflict resolution mechanism. If a tool fails, the system fails.
> * **RadAgents:** Introduces the **V-RAG verification loop**. We do not just "add" retrieval; we designed a mechanism where the Synthesizer uses retrieval as a grounding anchor to resolve conflicts between tools. This "Check-and-Balance" design is a novel contribution to trustworthy medical AI, going far beyond simple performance optimization.
>
> In summary, the **consistent ~12.3% performance gap** (73.6% vs 61.3% on ChestAgentBench) is not achievable through minor engineering tweaks. It stems from this structural evolution that addresses the fundamental limitations of single-agent systems in complex medical reasoning.

---

> > ### Author Response · Authors · 2026-01-24
> >
> > **(3) Will V-RAG introduce harmful retrievals?**
> >
> > Yes, at the input level, but it is effectively mitigated at the output level. Retrieval systems intrinsically introduce noise (e.g., ~14% harmful items in our MS-CXR pilot study, Fig. 7). However, our ChestAgentBench analysis confirms that the Synthesizer acts as a robust filter (**Table R1**):
> >
> > * **Input Risk:** Potential exposure to harmful retrieval items.
> > * **Output Impact:** The actual Negative Impact (where noise flips a correct answer to wrong) is suppressed to only 6.5%.
> >
> > **Table R1: Outcome Breakdown on ChestAgentBench (Subset n=866 where V-RAG was activated)**
> >
> > | Impact Category | Definition | Count | Rate | Analysis |
> > | --- | --- | --- | --- | --- |
> > | **Positive Impact** | (Wrong → Correct) | 224 | 25.9% | **Helpful:** Synthesizer successfully utilized the extra evidence to correct errors. |
> > | **Negative Impact** | (Correct → Wrong) | 56 | 6.5% | **Harmful:** Synthesizer was misled by retrieval noise. |
> > | **Neutral** | (Unchanged) | 586 | 67.6% | **Stable:** System maintained consistency. |
> > | **Net Gain** | Positive - Negative | +168 | +19.4% | Contributes to the overall +6.7% gain on the full set. |
> >
> > In summary, the system successfully filters out the majority of harmful retrievals via cross-modal reasoning, allowing the **Positive Impact (25.9%)** to drive a significant net performance gain.
> >
> > **(4) No reporting of inference latency or computational cost, despite system complexity.**
> >
> > We thank the reviewer for this practical question. We have conducted a detailed cost/latency analysis to demonstrate that the system is computationally feasible.
> >
> > **1. Mechanism Clarification (Sparse Activation):** First, we clarify that agents are sparsely activated. As described in Section 2.3, the Orchestrator activates only the specific sub-agents required for a query (e.g., a query about "bone fractures" will not activate the "Lung Opacity" agent). Therefore, we rarely incur the cost of running all 7 agents simultaneously. All idle agents incur zero computational overhead.
> >
> > **2. Benchmark Setup:** We measured performance on an on-premise node with 8$\times$NVIDIA RTX A5000 (24GB) using quantized models (Ollama backends) and LangGraph orchestration. We report: (1) **Latency:** End-to-end wall-clock time (seconds/case). (2) **Compute Cost:** Cumulative GPU usage (GPU-seconds/case).
> >
> > **Table R2: Latency and Cost Analysis**
> >
> > | Configuration | CheXbench (sec / GPU-sec) | ChestAgentBench (sec / GPU-sec) | Report Gen (sec / GPU-sec) |
> > | --- | --- | --- | --- |
> > | Single-Agent ReAct | 16 / 30 | 25 / 50 | 50 / 110 |
> > | Single-Agent Workflow | 18 / 36 | 29 / 64 | 58 / 140 |
> > | RadAgents (Sequential) | 38 / 89 | 62 / 178 | 120 / 368 |
> > | **RadAgents (Parallel)** | **26 / 83** | **41 / 164** | **85 / 340** |
> >
> > **Analysis:**
> >
> > * **Parallelism Efficiency:** Running sub-agents in parallel reduces wall-clock latency by ~30-40% compared to the sequential variant, making the system responsive enough for clinical workflows (e.g., ~41 seconds for complex ChestAgentBench queries).
> > * **Cost-Benefit Trade-off:** While RadAgents incurs roughly 2-3$\times$ the compute cost of a simple Single-Agent baseline, this is the necessary trade-off to achieve the 10%+ performance gains. In high-stakes medical diagnosis, where accuracy is paramount, we believe this increased computational cost is a justified investment.

---

### Official Review · Reviewer_BSzt · 2026-01-09

**Confidence:** 4
**Preliminary Rating:** 4
**Final Rating:** 4

**Summary:**

The authors present RadAgents, a multi-agent framework for Chest X-ray interpretation that explicitly models the clinical "ABCDE" (Airway, Breathing, Circulation, Diaphragm, Everything else) workflow. The system comprises an Orchestrator (for routing), five specialist Subagents (equipped with tools like segmentation, grounding, and VQA), and a Synthesizer that resolves inter-agent conflicts using Visual-RAG. By moving away from end-to-end "black box" reasoning to a structured, tool-augmented pipeline, the method achieves state-of-the-art performance on Chest Agent Bench and CheXbench, outperforming both specialist models (CheXagent) and generalist LLMs (GPT-4o).

**Strengths:**

- Clinical Alignment: The design of the agents around the standard ABCDE radiological assessment is a strong, clinically grounded choice. It imposes necessary structure on the reasoning process, preventing the "drift" often seen in unconstrained VLM generation.

- Traceability: Unlike end-to-end models, the multi-agent approach produces an audit trail of intermediate steps (e.g., "Subtask 1: segment heart," "Subtask 2: measure width"). This transparency is vital for trust in medical AI.

- Strong Empirical Results: The framework demonstrates impressive gains over strong baselines. Achieving 73.6% on Chest Agent Bench (vs. 39.5% for CheXagent) while using open-source models (Qwen-VL) suggests that architectural scaffolding is as important as model scale.

- Clarity and Completeness: The manuscript is well-written and easy to follow. The authors clearly articulate the architecture and provide sufficient detail regarding the tool definitions and agent interactions.

**Weaknesses:**

- System Complexity: The proposed framework is complex, involving 7 distinct agents and a vast suite of external tools (MAIRA-2, CXAS, BiomedParser, MedGemma, etc.). While effective, one might worry it is constructed from too many moving parts, while the robustness of this chain is not fully analyzed.

- Figures: Figure 1 could be explained better, especially with regard to difference of proposed approach of previous methods. The caption of Figure 3 could be enhanced to help the reader understand the method.

**Detailed Comments:**

- Figure 5: Consider reordering the bars to sort methods by ascending GREEN score.

- Figures 6 & 7:
- - Please explicitly link the discussion of these figures in the main text (e.g. Figure 7 is solely discussed in the Appendix).
- - Captions contain extra word 'figure'

**Justification Of Final Rating:**

The work remains methodologically strong and well-written. The authors provided a detailed rebuttal, including the requested latency analysis and a valuable ablation study. Thus, I maintain my positive score. of Weak Accept. The paper is a strong engineering contribution, but the novelty is primarily in the workflow alignment rather than the underlying methodology.

**Justification Of The Preliminary Rating:**

The work is methodologically strong and addresses a key limitation of medical VLMs (lack of structure) with excellent results. The manuscript is well-written and the clinical alignment is a major strength. The complexity of the pipeline raises concerns about robustness and deployment, whose discussion could be improved.

**Questions To Address In The Rebuttal:**

- Latency & Cost: You propose running 7 agents (potentially in parallel) and invoking multiple vision tools per query. What is the average inference time per case? Is this architecture feasible for real-time clinical workflows, or is it purely for offline analysis?

- Error Propagation: How does the system handle failures in upstream tools (e.g., if CXAS fails to segment the ribs)? Does the "Orchestrator" detect tool failure, or does the error cascade into the final report?

- Discussion & Limitations: What are current limiations of the approach? Where do you see future directions?

---

> ### Author Response · Authors · 2026-01-24
>
> Thank you for taking the time to review our paper and for providing such insightful comments. If our response addresses your concerns, we would greatly appreciate it if you could consider improving your review score.
>
> **(1) Latency & Cost: You propose running 7 agents (potentially in parallel) and invoking multiple vision tools per query. What is the average inference time per case? Is this architecture feasible for real-time clinical workflows, or is it purely for offline analysis?**
>
> We thank the reviewer for this practical question. We have conducted a detailed cost/latency analysis to demonstrate that the system is computationally feasible.
>
> **1. Mechanism Clarification (Sparse Activation):** First, we clarify that agents are sparsely activated. As described in Section 2.3, the Orchestrator activates only the specific sub-agents required for a query (e.g., a query about "bone fractures" will not activate the "Lung Opacity" agent). Therefore, we rarely incur the cost of running all 7 agents simultaneously. All idle agents incur zero computational overhead.
>
> **2. Benchmark Setup:** We measured performance on an on-premise node with 8$\times$NVIDIA RTX A5000 (24GB) using quantized models (Ollama backends) and LangGraph orchestration. We report: (1) **Latency (mean):** End-to-end wall-clock time (seconds/case). (2) **Compute Cost (mean):** Cumulative GPU usage (GPU-seconds/case).
>
> **Table R2: Latency and Cost Analysis**
>
> | Configuration | CheXbench (sec / GPU-sec) | ChestAgentBench (sec / GPU-sec) | Report Gen (sec / GPU-sec) |
> | --- | --- | --- | --- |
> | Single-Agent ReAct | 16 / 30 | 25 / 50 | 50 / 110 |
> | Single-Agent Workflow | 18 / 36 | 29 / 64 | 58 / 140 |
> | RadAgents (Sequential) | 38 / 89 | 62 / 178 | 120 / 368 |
> | **RadAgents (Parallel)** | **26 / 83** | **41 / 164** | **85 / 340** |
>
> **Analysis:**
>
> * **Parallelism Efficiency:** Running sub-agents in parallel reduces wall-clock latency by ~30-40% compared to the sequential variant, making the system responsive enough for clinical workflows (e.g., ~41 seconds for complex ChestAgentBench queries).
> * **Cost-Benefit Trade-off:** While RadAgents incurs roughly 2-3$\times$ the compute cost of a simple Single-Agent baseline, this is the necessary trade-off to achieve the 10%+ performance gains. In high-stakes medical diagnosis, where accuracy is paramount, we believe this increased computational cost is a justified investment.
>
> **Real-Time Feasibility:** With 26–41s latency (Parallel mode) on older GPUs (A5000) for easy-to-medium clinical tasks, RadAgents is immediately viable for asynchronous workflows (e.g., triage, report pre-population), fitting well within the standard acquisition-to-review window. Furthermore, scaling to modern enterprise hardware (e.g., A100/H100) would significantly accelerate inference, enabling true real-time deployment.
>
> **(2) Error Propagation: How does the system handle failures in upstream tools (e.g., if CXAS fails to segment the ribs)? Does the "Orchestrator" detect tool failure, or does the error cascade into the final report?**
>
> We address error propagation through the Synthesizer's weighted consensus mechanism, rather than explicit visual detection by the Orchestrator (which focuses on task planning). Failures are handled via three mechanisms:
>
> * **Confidence-Aware Weighing and Self-Check:** Tools can provide confidence scores, such as segmentation uncertainty or classification probabilities. The Synthesizer down-weights low-confidence outputs to prevent them from dominating the final report.
> * **Cross-Tool Redundancy:** If a specific tool (e.g., segmentation) fails, the Synthesizer relies on alternative evidence from global classification tools or V-RAG, ensuring the error does not cascade.
> * **V-RAG Grounding:** As shown in our ablation, V-RAG acts as a robust stabilizer, providing retrieval-based evidence to correct or override conflicting tool outputs.
>
> We also provide a demonstration in **Appendix E** that illustrates how our system manages failures of the upstream tools.

---

> > ### Author Response · Authors · 2026-01-24
> >
> > **(3) Discussion & Limitations: What are current limiations of the approach? Where do you see future directions?**
> >
> > We have expanded the Discussion section in the revised manuscript to address these points.
> >
> > **Current Limitations:**
> >
> > * **Dependency on Tool Capabilities:** The system's performance is upper-bounded by the quality of the underlying tools. Currently, most tools are optimized for frontal-view X-rays, limiting performance on lateral views. However, the modular design allows for seamless integration of improved tools as they become available.
> > * **Unidirectional Interaction:** The current Orchestrator can utilize tool outputs but cannot iteratively correct or guide them (e.g., refining a segmentation mask). The system handles errors via synthesis rather than active correction.
> >
> > **Future Directions:**
> >
> > * **Multi-Modality Extension:** Expanding the framework to support 3D modalities (e.g., CT, MRI) by integrating modality-specific sub-agents.
> > * **Efficiency Optimization:** While the multi-agent approach maximizes accuracy (critical for high-stakes medical diagnosis), it incurs higher computational costs. Future work will explore dynamic agent selection and context compression techniques to reduce latency without compromising performance.
> >
> > **(4) Figures: Figure 1 could be explained better, especially with regard to difference of proposed approach of previous methods. The caption of Figure 3 could be enhanced to help the reader understand the method.**
> >
> > We thank the reviewer for these helpful suggestions. We have revised the manuscript to explicitly contrast our approach with prior methods in Figure 1 and have expanded the caption of Figure 3 to better explain the synthesis mechanism.
> >
> > **1. Revisions regarding Figure 1 (Comparison with Prior Methods):**
> >
> > > **Figure 1 (revised).** Prior “encode-once, text-only” (left) and grounding/cropping-only variants (right) can fail on queries requiring iterative re-inspection and quantitative assessment (e.g., cardiothoracic ratio). RadAgents instead supports multimodal interleaved reasoning, where hypotheses trigger targeted visual operations (segmentation/measurement) and the final answer is grounded in explicit visual evidence.
> >
> > **2. Revised Caption for Figure 3:**
> > We have enhanced the caption to provide a step-by-step explanation of the V-RAG trigger mechanism:
> >
> > > **Figure 3 (updated caption).** **V-RAG Mechanism.** When tool outputs conflict, the system retrieves top- clinically similar CXR studies as reference standards to verify the findings and resolve the disagreement (e.g., Edema and Cardiomegaly).
> >
> > **(5) Figure 5: Consider reordering the bars to sort methods by ascending GREEN score. Figures 6 & 7: Please explicitly link the discussion of these figures in the main text (e.g. Figure 7 is solely discussed in the Appendix). Captions contain extra word 'figure'**
> >
> > Thank you for these helpful formatting suggestions. We have incorporated them as follows:
> >
> > * **Figure 5:** We have reordered the bars by ascending GREEN score to improve readability.
> > * **Figures 6 & 7:** We have corrected the caption typos. Figure 7 is explicitly referenced in Section 3.1, with its detailed analysis provided in the Appendix due to page limits.

---

### Official Review · Reviewer_uQCP · 2026-01-10

**Confidence:** 4
**Preliminary Rating:** 3
**Final Rating:** 4

**Summary:**

RadAgents is a multi-agent framework for chest X-ray interpretation. It structures reasoning around the clinical (systematic) ABCDE review scheme. The system has an orchestrator (for task dispatch), five specialized sub-agents, and a synthesizer that integrates outputs from the sub-agents and resolves conflicts by comparing with similar retrieved images from the database. I see the key claim as encoding radiologist-style workflows into agents, along with tool augmentation and image retrieval, achieves more accurate and interpretable analysis compared to (a) end-to-end and (b) single-agent approaches. The authors have evaluated the system on three benchmarks and have shown consistent improvements over the baselines along with ablations for workflow orchestration and retrieval.

**Strengths:**

- I believe the central strength of the paper is the ABCDE decomposition. It is clinically sensible and mirrors how radiology trainees actually learn systematic X-ray review.
- This design also allows the model to provide step-level traceability, i.e. explicit intermediate artifacts including anatomy/pathology segmentations, measurements, similar retrieved images, and rationales rather than post-hoc explanations / text rationales.
- The separation between workflows, skills and tools is practical engineering choice and well though out. It allows for dynamic tool composition for each of the ABCDE agent.
- The system also shows strong results with an 8B (easily deployable) open model.
- The V-RAG ablation in figure 7 honestly reports the helpful-vs-harmful tradeoff. Many papers would hide the harmful rate.
- Ablations in figure 6 are targeted and informative, e.g. model scaling vs. task dispatch for conflict resolution for where to invest the compute in.
- Overall, I consider this as a systems paper that is carefully engineered and empirically solid.

**Weaknesses:**

- Ablations show multi-agent beats single-agent. It doesn't isolate whether ABCDE structure itself matters. An alternate multi-agent decomposition (e.g. upper/mid/lower thorax or generic/random splits) might achieve similar gains. Is the gain coming from clinical alignment? Or simply from multi-agent parallelism (with conflict resolution)?
- Figure 6 shows even with a 30B synthesizer, the conflict resolution rate reaches only 60%. This means roughly 40% of the tool disagreements remain unsolved. What happens in these cases? Does the system default to one tool? Abstain?
- Only GREEN score (LLM-as-a-judge metric) for report generation is used. Standard metrics (RadGraph F1/CheXbert F1 or clinical error categorization or other entity-level metrics) are not reported (and is important to have confidence in the clinical accuracy claims).
- Figure 7 shows 14% harmful retrievals at k=3 (yet V-RAG ablation shows a 6-7 point gain). Is V-RAG helping because of good retrievals or despite the harmful ones? Does the synthesizer effectively filter the harmful retrievals? Breaking down accuracy by helpful/harmful cases would help.
- V-RAG retrieval corpus is not described. Key questions: what dataset populates the index? Are test patients excluded entirely (not just test images)? Without this information, 6-7 point gain from V-RAG integration is hard to interpret.

**Detailed Comments:**

- Section 2.1 could justify tool choices (why CXAS over alternatives, why MedGemma for VQA).
- Paper doesn't report frequency of workflow-free ReAct fallback or SkillMismatchError.
- Appendix provides skill set examples. But full system prompts, tool configurations and complete workflow specifications are not included. For a systems paper of this nature, releasing these (or codebase) would help future work benchmark properly against RadAgents.
- Can you provide cost/latency analysis? Running 7 agents in parallel along with multiple tool calls per query is computationally expensive.
- Paper claims step-level traceability but does not show a complete trace even for a single case. An example showing the full pipeline (orchestrator dispatch, subagent reasoning, tool calls, conflict detection + resolution) in the appendix would strengthen the claim.

**Justification Of Final Rating:**

The rebuttal addresses my major concerns. The ABCDE ablation shows clinical decomposition contributes modestly (~1.9%) over generic splits. This is an honest result that tempers the original framing but doesn't undermine the work. The real contribution is the multi-agent architecture with conflict resolution and V-RAG verification, which the ablations support. Entity-level metrics now included. The system is well-engineered and the evaluation is now thorough. Main remaining concern is silent failure on unresolved conflicts (no uncertainty flagging).

**Justification Of The Preliminary Rating:**

The system is well engineered. The evaluation is thorough. Multi-agent beats single-agent consistently across three benchmarks. But I believe the main claim is that ABCDE alignment structure is what makes it work. This is not tested. Would any five agent split do similarly well? The gains could be from a) parallelism, b) reduced context per agent or c) conflict resolution rather than clinical alignment? V-RAG helps but the retrieval corpus details are missing. Overall, this is a solid engineering on top of existing components but the core claim (that clinical alignment helps the gains) needs direct ablation support.

**Questions To Address In The Rebuttal:**

- Have you compared the ABCDE decomposition against alternative multi-agent structures (region based or generic/random splits)? Discuss if the gains come from workflow alignment or from multi-agent parallelism itself?
- For the ~40% of tool conflicts that remain unresolved, what is the system behaviour? Default? Abstain? Flagged?
- For V-RAG, can you show accuracy breakdown for cases with helpful vs. harmful retrievals?
- Will you release full system prompts, tool configurations and complete workflow specifications (or codebase) for reproducibility? This would help future work benchmark against RadAgents properly.
- Can you show atleast one detailed trace showing the full pipeline to strengthen the step-level traceability claim?

---

> ### Author Response · Authors · 2026-01-24
>
> Thank you for taking the time to review our paper and for providing thoughtful comments. If our response addresses your concerns, we would greatly appreciate it if you could consider improving your review score.
>
> **(1) Have you compared the ABCDE decomposition against alternative multi-agent structures? Discuss if the gains come from workflow alignment or from multi-agent parallelism itself?**
>
> Thank you for the suggestion. We add a controlled ablation with V-RAG disabled. The only change is how sub-agent roles are defined and how the orchestrator-produced sub-tasks are assigned:
>
> * **ABCDE-5 (ours):** Five role-specific agents with the original ABCDE prompts.
> * **Generalist-5 (balanced random assignment):** Five agents share an identical generalist prompt (no ABCDE specialization); the orchestrator generates the same set of sub-tasks, which are assigned to agents using a balanced random mapping, with no replacement per case (to prevent all tasks from accumulating on a single agent).
>
> | Setting | ChestAgentBench Acc | CheXbench Acc | MIMIC GREEN |
> | --- | --- | --- | --- |
> | **ABCDE-5** | **66.9%** | **71.5%** | **46.1** |
> | Generalist-5 | 65.0% | 71.4% | 43.7 |
>
> This ablation directly shows that under identical orchestration, conflict resolution, tool access, and inference budgets, removing ABCDE role specialization (Generalist-5) leads to consistent degradation on ChestAgentBench (**−1.9%** absolute) and MIMIC-CXR report generation (**−2.4** GREEN), while the gap on CheXbench is negligible (−0.1%). This pattern is consistent with the dataset characteristics: CheXbench is closer to single-shot reasoning where parallelism and synthesis alone can already perform well, whereas ChestAgentBench and MIMIC-CXR require more structured tool-mediated evidence and multi-step integration, where clinically aligned decomposition provides additional benefit beyond parallelism.
>
> Beyond overall scores, ABCDE offers two practical system advantages that our generalist baseline lacks:
>
> 1. **Traceability and debugging:** Responsibilities are explicit, so failures can be localized to a specific subsystem agent (e.g., airway vs circulation) rather than being diffused across interchangeable generalists.
> 2. **Controlled context and integration:** Role-scoped intermediate outputs make synthesis more structured (less redundant overlap across agents), which is particularly important for report generation where long-form context integration is a bottleneck.
>
> Taken together, the results indicate that ABCDE contributes not only to accuracy on workflow-heavy tasks but also to a more interpretable and maintainable multi-agent pipeline, whereas a role-agnostic random assignment mainly preserves generic parallelism without these benefits.
>
> **(2) For the ~40% of tool conflicts that remain unresolved, what is the system behaviour? Default? Abstain? Flagged?**
>
> We thank the reviewer for this careful observation. To clarify, the **"Conflict Resolution Rate"** is a correctness metric (did the system output the factually correct diagnosis?), not a completion metric. Therefore, the "unsolved" 40% represents cases where the system generated an **incorrect answer**, rather than crashing or timing out.
>
> **System Behavior in Unsolved Cases:** In these instances, the system does not abstain (i.e., it does not output "I don't know") nor does it have a hard-coded default (e.g., "always trust Tool A"). Instead, it attempts to synthesize a conclusion but fails to identify the correct evidence. This typically happens for two reasons:
>
> * **Model Capacity:** The Synthesizer's multimodal in-context learning capabilities are bounded. In complex scenarios, it may fail to correctly weigh the conflicting evidence (e.g., trusting a confident but erroneous text description over a correct numerical measurement).
> * **V-RAG Noise:** As analyzed in Figure 7, retrieving neighbors can sometimes introduce "harmful" context. If the retrieved examples are visually similar but differ in diagnosis, they can mislead the Synthesizer into hallucinating a match where none exists.
>
> **(3) Only GREEN score (LLM-as-a-judge metric) for report generation is used. Standard metrics (RadGraph F1/CheXbert F1 or clinical error categorization or other entity-level metrics) are not reported.**
>
> Thank you for the suggestion. We use GREEN as the primary metric to ensure readers are not overwhelmed with too many metrics. We agree that other standard metrics are very important, and the corresponding results are listed below. This content is also updated in the updated manuscript (**Appendix F.1**).
>
> | Settings / Metrics | CheXbert-macro-F1 (14) | RadGraph-F1 | RaTE |
> | --- | --- | --- | --- |
> | CheXagent | 29.2 | 12.7 | 44.4 |
> | GPT-4o | 22.4 | 15.3 | 49.8 |
> | Qwen3-VL w/ ReAct | 44.9 | 17.1 | 53.6 |
> | Qwen3-VL w/ Workflow | 45.5 | 17.2 | 53.9 |
> | RadAgents w/o V-RAG | 49.1 | 18.1 | 55.8 |
> | **RadAgents** | **53.2** | **19.4** | **58.5** |

---

> > ### Author Response · Authors · 2026-01-24
> >
> > **(4) Figure 7 shows 14% harmful retrievals at k=3 (yet V-RAG ablation shows a 6-7 point gain). Is V-RAG helping because of good retrievals or despite the harmful ones? Does the synthesizer effectively filter the harmful retrievals? Breaking down accuracy by helpful/harmful cases would help.**
> >
> > We thank the reviewer for this insightful suggestion. We agree that understanding the interplay between retrieval quality and model robustness is critical.
> >
> > **1. Clarification on Data and Granularity:** We emphasize two distinctions:
> >
> > * **Dataset:** Figure 7 analyzes the MS-CXR subset (pilot study for k-selection). However, the main accuracy gains (6-7 points) reported in the paper are based on ChestAgentBench. Therefore, the following breakdown focuses on ChestAgentBench to align with the main results.
> > * **Item vs. Query:** The system's final performance depends on the Query-level impact (how the Synthesizer processes the set of top-3 examples as a holistic context), rather than just the item-level relevance.
> >
> > **2. Breakdown of Accuracy (Filtering Analysis):** To directly answer *"Does the synthesizer filter harmful retrievals?"*, we analyzed the post-hoc outcome changes on the ChestAgentBench subset where V-RAG was activated (n=866) in RadAgents. We classified the impact based on whether the Top-3 context caused the final answer to flip:
> >
> > **Table R1: Outcome Breakdown on ChestAgentBench (Subset n=866)**
> >
> > | Impact Category | Definition | Count | Rate | Analysis |
> > | --- | --- | --- | --- | --- |
> > | **Positive Impact** | (Wrong →  Correct) | 224 | 25.9% | **Helpful:** Synthesizer successfully utilized the extra evidence to correct errors. |
> > | **Negative Impact** | (Correct → Wrong) | 56 | 6.5% | **Harmful:** Synthesizer was misled by retrieval noise. |
> > | **Neutral** | (Unchanged) | 586 | 67.6% | **Stable:** System maintained consistency. |
> > | **Net Gain** | Positive - Negative | +168 | +19.4% | Contributes to the overall +6.7% gain on the full set. |
> >
> > **Conclusion:**
> >
> > * **Is V-RAG helping because of good retrievals?** Yes. The 25.9% Positive Impact is the primary driver of the performance gain. This represents cases where the retrieved context successfully corrected baseline errors.
> > * **Does it help despite the harmful ones?** Yes (Filtering confirmed). The analysis shows a **4:1 Benefit-to-Cost Ratio** (25.9% vs. 6.5%). Although retrieval noise is inevitable in any RAG system, the Synthesizer suppresses the actual Negative Impact to only 6.5%. The fact that the positive corrections outweigh the negative flips by a factor of four confirms that the Synthesizer effectively filters noise and prioritizes helpful signals to achieve a robust net gain.
> >
> > **(5) V-RAG retrieval corpus is not described.**
> >
> > The retrieval index consists exclusively of the MIMIC-CXR official training split. We strictly enforce patient-level exclusion (via Subject_ID), ensuring test patients are completely disjoint from the retrieval corpus. For external benchmarks (e.g., ChestAgentBench), the data originates from distinct sources, ensuring zero overlap with the MIMIC-CXR index.
> >
> > **(6) Section 2.1 could justify tool choices (why CXAS over alternatives, why MedGemma for VQA).**
> >
> > Thanks for the suggestion. We choose ROI/grounding/VQA tools to (i) provide verifiable intermediate evidence (masks/boxes/measurements), (ii) maximize CXR-relevant coverage for our workflows, and (iii) ensure reproducibility with open-sourced checkpoints.
> > In particular, CXAS is selected for broad CXR anatomy coverage needed by downstream measurement pipelines, and MedGemma is used as a flexible VQA specialist due to its strong ability to follow instructions for diverse clinical queries.
> >
> > **(7) Paper doesn't report frequency of workflow-free ReAct fallback or SkillMismatchError.**
> >
> > Thank you for the suggestion. The average frequencies as follows:
> >
> > | Datasets                | Workflow-free (ReAct) fallback rate | SkillMismatchError / re-dispatch rate |
> > | ------------------------ | -----------------------------------------: | -------------------------------------------: |
> > | CheXbench**            |                                   10.4% |                                      5.3% |
> > | ChestAgentBench**      |                                   18.1% |                                     9.3% |
> > | MIMIC-CXR Report Gen |                                    2.1% |                                      4.0% |
> >
> > Interpretation: ChestAgentBench contains more open-ended, mixed-intent queries, hence a higher fallback and re-dispatch rate; MIMIC-CXR report generation is closer to a fixed workflow, hence the lowest workflow-free fallback; CheXbench falls in between.
> >
> > **(8) Will you release full system prompts, tool configurations and complete workflow specifications (or codebase) for reproducibility?**
> >
> > Yes, we plan to release the code after obtaining organizational approval and upon publication.

---

> > > ### Author Response · Authors · 2026-01-24
> > >
> > > **(9) Can you provide cost/latency analysis? Running 7 agents in parallel along with multiple tool calls per query is computationally expensive.**
> > >
> > > We thank the reviewer for this practical question. We have conducted a detailed cost/latency analysis to demonstrate that the system is computationally feasible.
> > >
> > > **1. Mechanism Clarification (Sparse Activation):** First, we clarify that agents are sparsely activated. As described in Section 2.3, the Orchestrator activates only the specific sub-agents required for a query (e.g., a query about "bone fractures" will not activate the "Lung Opacity" agent). Therefore, we rarely incur the cost of running all 7 agents simultaneously. All idle agents incur zero computational overhead.
> > >
> > > **2. Benchmark Setup:** We measured performance on an on-premise node with 8$\times$NVIDIA RTX A5000 (24GB) using quantized models (Ollama backends) and LangGraph orchestration. We report: (1) **Latency:** End-to-end wall-clock time (seconds/case). (2) **Compute Cost:** Cumulative GPU usage (GPU-seconds/case).
> > >
> > > **Table R2: Latency and Cost Analysis**
> > >
> > > | Configuration | CheXbench (sec / GPU-sec) | ChestAgentBench (sec / GPU-sec) | Report Gen (sec / GPU-sec) |
> > > | --- | --- | --- | --- |
> > > | Single-Agent ReAct | 16 / 30 | 25 / 50 | 50 / 110 |
> > > | Single-Agent Workflow | 18 / 36 | 29 / 64 | 58 / 140 |
> > > | RadAgents (Sequential) | 38 / 89 | 62 / 178 | 120 / 368 |
> > > | **RadAgents (Parallel)** | **26 / 83** | **41 / 164** | **85 / 340** |
> > >
> > > **Analysis:**
> > >
> > > * **Parallelism Efficiency:** Running sub-agents in parallel reduces wall-clock latency by ~30-40% compared to the sequential variant, making the system responsive enough for clinical workflows (e.g., ~41 seconds for complex ChestAgentBench queries).
> > > * **Cost-Benefit Trade-off:** While RadAgents incurs roughly 2-3$\times$ the compute cost of a simple Single-Agent baseline, this is the necessary trade-off to achieve the 10%+ performance gains. In high-stakes medical diagnosis, where accuracy is paramount, we believe this increased computational cost is a justified investment.
> > >
> > > **(10) Can you show atleast one detailed trace showing the full pipeline to strengthen the step-level traceability claim?**
> > >
> > > We thank the reviewer for the suggestion. We have added a comprehensive step-level execution trace in **Appendix E** of the revised manuscript.
> > >
> > > This detailed case study (query: *"Is the trachea midline?"*) explicitly visualizes the full pipeline:
> > >
> > > * **Orchestrator Dispatch:** Routing to the specific Airway Agent.
> > > * **Tool Execution:** Capturing a tool's heuristic error.
> > > * **Conflict Detection & Resolution:** Demonstrating how the Synthesizer detects the discrepancy and triggers **V-RAG** to correct the diagnosis.
> > >
> > > This provides the concrete "white-box" evidence requested to substantiate our traceability claims.
> > >
> > > **(11) I believe the main claim is that ABCDE alignment structure is what makes it work.**
> > >
> > > We appreciate the reviewer’s careful reading. We would like to clarify that our main claim is not that ABCDE is uniquely optimal, but that is a traceable, tool-augmented, verification-aware agentic framework (which can integrate with open-sourced model as the control core) that can produce visually grounded evidence and explicit reasoning trajectories and resolve cross-tool inconsistencies, which addresses limitations of prior tool-aggregation pipelines. In the following we will address your concerns point-by-point.

---

### Author Rebuttal · Authors · 2026-01-24

**Rebuttal:**

We sincerely thank the reviewers for their constructive feedback and for recognizing our work as "clinically sensible" and "empirically solid". RadAgents represents a shift from sequential tool aggregation to a collaborative, verification-aware agentic framework. Unlike prior linear pipelines (e.g., MedRAX), our system introduces novel and clinically-significant architectural innovations.

We have incorporated new analyses and data requested by the reviewers. All changes are marked in **red** in the revised PDF:

* **Step-Level Traceability (Appendix E):** Added a comprehensive execution trace (Query: "Is the trachea midline?") visualizing the full pipeline: Orchestrator dispatch, Tool execution, and Conflict Resolution via V-RAG .


* **Cost & Latency Analysis (Appendix F.2):** Included a detailed breakdown of inference latency and computational cost, demonstrating the efficiency gains of the Parallel architecture (Table 4) .


* **Additional Standard Metrics (Appendix F.1):** Reported CheXbert-F1, RadGraph-F1, and RaTEScore comparisons as requested (Table 3) .


* **Systematic Error Analysis (Appendix F.3):** Added an analysis of workflow fallback rates and internal dispatch errors across datasets (Table 5) .


* **V-RAG Details (Appendix D):** Clarified the retrieval corpus construction (strict patient-level exclusion) and added a sensitivity analysis for  selection (Figure 7 analysis) .


* **Discussion & Limitations (Section 4):** Expanded the discussion to explicitly address tool dependency, unidirectional interaction limits, and future efficiency optimizations .


* **Figure Updates:** Revised Figure 1 caption to contrast with "encode-once" methods and Figure 3 caption to clarify the V-RAG trigger mechanism .

**Supporting Material:**

/attachment/aecabbd678e39de5c93db26eaf7da27675050c66.pdf

---

### Comment · Area_Chair_dJec · 2026-02-01
**Please enter Final Ratings**

Dear reviewers,
Please note that today, Feb 1 is the last day to enter your final ratings. Thank you to those who have already updated. If you have not yet, please take a moment to look through the author’s rebuttal and update your final score and reasoning.
We greatly appreciate your important contribution to MIDL.
Thank you!
Your AC

---

### Meta-Review · Area_Chair_dJec · 2026-02-09

**Recommendation:** Accept (Poster)
**Confidence:** 4

**Metareview:**

Reviewers final ratings are somewhat mixed, although trend positively (2 WA, 1 borderline increased from weak reject post rebuttal). The greatest concerns are regarding the novelty of methodology and perception of missing comparative baselines. Still, reviewers agree that the work presents a solid engineering contribution, proposing alignment of agents with radiology workflow and multi-agent conflict resolution, and the experiments show strong empirical results. Given the timeliness of this line of research, I think there will be strong interest in the MIDL community and recommend acceptance of this paper.

---

### Decision · Program_Chairs · 2026-02-13

Accept (Poster)